# Neutrophil-to-Lymphocyte, Platelet-to-Lymphocyte and Monocyte-to-Lymphocyte Ratio in Bipolar Disorder

**DOI:** 10.3390/brainsci11010058

**Published:** 2021-01-06

**Authors:** Laura Fusar-Poli, Antimo Natale, Andrea Amerio, Patriciu Cimpoesu, Pietro Grimaldi Filioli, Eugenio Aguglia, Mario Amore, Gianluca Serafini, Andrea Aguglia

**Affiliations:** 1Psychiatry Unit, Department of Clinical and Experimental Medicine, University of Catania, 95124 Catania, Italy; laura.fusarpoli@gmail.com (L.F.-P.); antimo.natale88@gmail.com (A.N.); eugenio.aguglia@unict.it (E.A.); 2Department of Neuroscience, Rehabilitation, Ophthalmology, Genetics, Maternal and Child Health, Section of Psychiatry, University of Genoa, 16126 Genoa, Italy; andrea.amerio@unige.it (A.A.); patriciu.cimpoesu@gmail.com (P.C.); piotr.grimaldi@gmail.com (P.G.F.); mario.amore@unige.it (M.A.); gianluca.serafini@unige.it (G.S.); 3IRCCS Ospedale Policlinico San Martino, 16132 Genoa, Italy; 4Department of Psychiatry, Tufts University, Boston, MA 02110, USA

**Keywords:** inflammatory ratio, platelet, monocyte, lymphocyte, blood cell counts, inflammation, biomarker, mania

## Abstract

Background: Several inflammatory hypotheses have been suggested to explain the etiopathogenesis of bipolar disorder (BD) and its different phases. Neutrophil-to-lymphocyte (NLR), platelet-to-lymphocyte (PLR), and monocyte-to-lymphocyte (MLR) ratios have been proposed as potential peripheral biomarkers of mood episodes. Methods: We recruited 294 patients affected by BD, of which 143 were experiencing a (hypo)manic episode and 151 were in a depressive phase. A blood sample was drawn to perform a complete blood count. NLR, PLR, and MLR were subsequently calculated. A *t*-test was performed to evaluate differences in blood cell counts between depressed and (hypo)manic patients and a regression model was then computed. Results: Mean values of neutrophils, platelets, mean platelet volume, NLR, PLR, and MLR were significantly higher in (hypo)manic than depressed individuals. Logistic regression showed that PLR may represent an independent predictor of (hypo)mania. Conclusions: Altered inflammatory indexes, particularly PLR, may explain the onset and recurrence of (hypo)manic episodes in patients with BD. As inflammatory ratios represent economical and accessible markers of inflammation, further studies should be implemented to better elucidate their role as peripheral biomarkers of BD mood episodes.

## 1. Introduction

Bipolar disorder (BD) is a chronic psychiatric disorder characterized by the alternation of different phases, specifically hypomanic/manic and major depressive episodes. BD affects 2–3% of the general population worldwide and its onset typically occurs during adolescence or early adulthood [1]. The Fifth Edition of the Diagnostic and Statistical Manual of Mental Disorders (DSM-5) includes three main types of BD in the category of “Bipolar and related disorders”: BD type I, BD type II, and cyclothymic disorder [2]. Patients with BD type I have experienced at least one manic episode (defined as an abnormally elevated mood or irritability and related symptoms with severe functional impairment or psychotic symptoms for seven days or more), with or without major depressive episodes. Patients with BD type II have experienced at least one hypomanic episode (defined as abnormally elevated mood or irritability and related symptoms with decreased or increased function for four days) and one major depressive episode [2].

BD is a heterogeneous and multifactorial disease, caused by a combination of genetic and environmental factors. Over recent years, there has been a growing interest towards the role of the immune system and inflammatory pathways in the etiology of BD [3,4]. Indeed, immunity and inflammatory imbalance may affect mental functioning from the earliest stages of neural development [5,6]. This notion is supported by the fact that various medical conditions characterized by chronic inflammation are associated with the onset and progression of BD [7]. Additionally, adjunctive anti-inflammatory agents may improve BD symptomatology during different mood episodes [8,9,10]. For instance, N-acetylcysteine, pioglitazone, minocycline, and coenzyme Q10 have demonstrated promising antidepressant effects. Contrarywise, even if with limited evidence, celecoxib (i.e., a nonsteroidal anti-inflammatory drug) may exert an antimanic effect [8].

A recent meta-analysis evidenced significantly higher levels of chemokines (i.e., a subgroup of cytokines that can induce directed chemotaxis to the sites of inflammation or injuries) in BD patients than controls [11]. Interestingly, the increased levels of chemokines also persisted during the euthymic phase, suggesting that these inflammatory mediators could represent longstanding markers of the disorder [12]. Another systematic review displayed a reduction in anti-inflammatory cytokines and an increase in pro-inflammatory cytokines in BD patients, particularly during manic and depressive phases, further supporting the association between inflammation and BD acute phases [13]. Of note, alterations in the inflammatory pathways are not limited to peripheral markers, as the presence of brain excitotoxicity and inflammation in patients with BD has been detected both in vivo [14] and post mortem [15].

Neutrophil-to-lymphocyte ratio (NLR), platelet-to-lymphocyte ratio (PLR), and monocyte-to-lymphocyte ratio (MLR) have been recently proposed as inflammatory markers. In fact, they represent low-cost and reproducible tests that can be easily calculated from a blood sample collected under simple laboratory conditions. These parameters appear associated with mood disorders, supporting the inflammatory hypothesis underlying the etiopathogenesis of these conditions [16]. Up to the present date, several studies have examined the usefulness of NLR, PLR, and MLR as potential biomarkers of BD. For instance, Çakir et al. [17] reported that NLR was significantly more elevated in patients with BD than healthy controls. Other studies were specifically focused on specific mood episodes, showing increased NLR and PLR values [18], as well as decreased MLR values [19] in manic patients than healthy subjects. Kalelioglu et al. instead showed increased NLR and PLR levels in BD patients than controls, but no differences between euthymic and manic phases [20]. Interestingly, elevated NLR in bipolar patients has also been associated with a reduction in cognitive functions, particularly attention [21]. Moreover, a recent study proposed an association between lethality of suicide attempts and different biologic parameters, including hematological parameters. According to the authors, higher mean platelet volume and PLR characterized the group of high-lethality suicide attempters [22]. Similarly, Ekinci et al. estimated the blood cell inflammatory ratios in depressed patients with and without suicide attempts, suggesting that NLR may be a trait marker for suicidal vulnerability in patients diagnosed with depression [23]. In fact, even if many studies regarding the association between suicidal behaviors and neuroinflammation have been conducted, a causal link between the two conditions still has to be proven [24].

A meta-analysis by Mazza et al. [25] summarized the literature on the topic, reporting that subjects with BD had higher NLR and PLR compared to healthy controls. Particularly, subgroup analyses evidenced that patients in manic and any bipolar phase showed significantly higher NLR and PLR than controls. However, the significance was lost while considering the subgroup of studies which recruited only euthymic bipolar subjects. Interestingly, only one study included in the review evaluated MLR [26], finding higher values in bipolar patients than controls. More recently, a cross-sectional investigation by Mazza et al. for the first time compared blood cell inflammation markers between different mood episodes of BD. The authors found higher NLR and MLR in bipolar manic than depressive episodes, suggesting that inflammatory changes may occur especially during acute episodes of mania [27].

Given the inflammatory mechanisms involved in the onset and recurrence of BD mood episodes, and the recent literature findings, the present study aimed to evaluate the differences in blood cell counts and inflammatory ratios, particularly NLR, PLR, and MLR, during the different mood episodes in patients affected by BD (i.e., depression and (hypo)mania).

## 2. Materials and Methods

### 2.1. Sample

The study had a cross-sectional design and involved inpatients with a primary diagnosis of bipolar disorder who were consecutively referred to the Section of Psychiatry, Department of Neuroscience, Rehabilitation, Ophthalmology, Genetics, Maternal and Child Health (DINOGMI), IRCCS Ospedale Policlinico San Martino, University of Genoa (Italy) and the Psychiatry Unit, Department of Clinical and Experimental Medicine, Gaspare Rodolico Hospital, University of Catania (Italy), over a period of three years (January 2017 to December 2019).

Inclusion criteria consisted of: (a) hospitalization in a psychiatric unit with a primary diagnosis of bipolar disorder for a current (hypo)manic or major depressive episode; (b) 18 years of age or older; (c) written informed consent to participate in the study. Exclusion criteria were: (a) pregnancy or recent childbirth, (b) severe and/or acute medical comorbidities or any other conditions that could affect the measured parameters, (c) clozapine prescription within the past six months, (d) current substance use disorder, including alcohol, (e) a positive history of acute neurological injury, and (f) the inability or refusal to provide a written informed consent to participate in the study.

Potential participants were given a thorough explanation of the study’s aims and procedures and the opportunity to ask questions. Written informed consent was obtained from all participants according to the current version of the Declaration of Helsinki. The study design was reviewed and approved by the “IRCCS Ospedale Policlinico San Martino” Ethical Review Board (82/13, amended on 10 February 2017, and registered with number 028 of 2 March 2017).

### 2.2. Assessments and Procedures

Basic sociodemographic and clinical features were obtained through the administration of a semi-structured interview including age, gender, marital and occupational status, educational level, psychiatric diagnosis, and pharmacological treatment. Psychiatric diagnoses were based on the Diagnostic and Statistical Manual of Mental Disorders, fifth edition (DSM-5) criteria and formulated by expert psychiatrists in inpatient clinical setting [2].

A blood test was performed to evaluate red blood cell (RBC) count, hemoglobin, hematocrit, mean corpuscular volume (MCV), mean corpuscular hemoglobin (MCH), mean corpuscular hemoglobin concentration (MCHC), red blood cell distribution width-coefficient of variation (RDW-CV), neutrophils, lymphocytes, monocytes, eosinophils and basophils, and platelets. Blood samples were taken in the morning (between 7 and 9 a.m.) of the first day of hospitalization, after 12 h of fasting, from a forearm vein. For each patient, about 3 mL of blood was collected in hemogram tubes containing EDTA. Samples were processed within 30 min after collection with flow cytometry in the respective laboratory sections (IRCCS Ospedale Policlinico San Martino, Genoa, and Policlinico “G. Rodolico”, Catania).

### 2.3. Statistical Analysis

Continuous and categorical variables were presented as means and standard deviations (SD) or frequency and percentage, respectively. Normal distribution was assessed using the Kolmogorov-Smirnov test, before applying statistical analyses.

The sample was divided into two subgroups according to bipolar illness phase: (a) patients experiencing a (hypo)manic episode and (b) patients experiencing a current major depressive episode. Student’s *t*-test for independent samples was performed to evaluate differences between mood episodes. We calculated effect sizes for the *t*-test (Cohen’s *d*) and interpreted them as follows: 0.2 was considered a small effect size, 0.5 a medium effect size, and 0.8 a large effect size. A binary logistic regression analysis was used to explore the relationship between bipolar patients experiencing a (hypo)manic episode and each of the significant independent variables found in the *t*-test, correcting for age and gender.

The Statistical Package for Social Sciences (SPSS) for Windows 25.0 (IBM Corp., Armonk, NY, USA) was used to carry out all the mentioned statistical analyses and the value of statistical significance was set at *p* < 0.05 (two tailed).

## 3. Results

### 3.1. General Characteristics of the Sample

Two-hundred ninety-four bipolar inpatients were enrolled in the present study, of which 151 (51.4%) were experiencing a current major depressive episode and 143 (48.6%) a current (hypo)manic episode. The mean (±SD) age of the total sample was 51.57 (±13.64). One-hundred fifty-one (53.4%) were females, and 32.3% (*N* = 95) were employed. As for the pharmacological treatment, patients were taking on average 3.81 (±1.22) medications and most patients were taking antipsychotics (*N* = 202, 85.6%). Other sociodemographic and clinical characteristics are displayed in Table 1.

### 3.2. Complete Blood Count and Inflammatory Ratios: Differences between Major Depressive and (Hypo)manic Episode

As regards complete blood count, bipolar patients experiencing a current (hypo)manic episode had a higher number of neutrophils (4.87 ± 1.89 vs. 4.37 ± 1.56, *p* = 0.014) and platelets (261.25 ± 68.80 vs. 234.73 ± 58.97, *p* < 0.001) compared to bipolar patients with a current major depressive episode. Furthermore, NLR (2.46 ± 1.33 vs. 1.89 ± 1.01, *p* < 0.001), PLR (130.13 ± 52.84 vs. 106.05 ± 36.79, *p* < 0.001), and MLR (0.28 ± 0.18 vs. 0.23 ± 0.12, *p* = 0.008) were significantly higher in the (hypo)manic than depressed group. Specifically, effect size was medium for NLR and PLR (Cohen’s *d* = 0.41 and 0.48, respectively), and small-to-medium for MLR (*d* = 0.33). Other non-statistically significant differences between the two identified subgroups were summarized in Table 2.

The means and standard deviations of the three inflammatory ratios of interest (NLR, PLR, and MLR) in the two groups are presented in Figure 1.

### 3.3. Independent Predictors of (Hypo)manic Episodes

Finally, we computed a binary logistic regression model using the phases of bipolar illness (0 = depression, 1 = mania) as dependent variables, and the values resulted significant in the *t*-tests as independent variables. The model was marginally significant (*R*^2^ Nagelkerke = 0.32, *p* = 0.05). In the regression model, after correcting for age and sex, only PLR (OR = 1.12, *p* = 0.03) was significantly associated with (hypo)manic illness phase among biological parameters (Table 3).

## 4. Discussion

The inflammatory system plays an important role in the onset and progression of mood disorders, and several studies have been conducted on the topic. The present study aimed to replicate the findings of Mazza et al. [27] who first attempted to evaluate complete blood count as a potential biomarkers of different mood episodes of BD. Indeed, we found that neutrophils, platelets, PLR, NLR, and MLR were significantly higher in patients experiencing a manic episode than depressed individuals. However, after performing a multiple logistic regression, only PLR was regarded as an independent predictor of a manic episode.

Our findings are in line with Mazza et al. [27], showing that inflammatory changes occur more frequently during hypo(manic) phases of BD than during major depressive episodes. Moreover, they are consistent with previous studies in which manic patients had higher levels of pro-inflammatory cytokines, such as C-reactive protein (CRP), soluble interleukin (IL)-6 receptor, and soluble tumor necrosis alpha (TNF-α) receptor, as compared to patients with major depressive episodes, suggesting more severe inflammatory dysregulation in mania [28]. Pro-inflammatory cytokines appear to be a state marker, as they are increased in manic episodes but not in a euthymic state [29,30,31]. Moreover, their levels seem to be responsive to lithium therapy [32,33]. Of note, pro-inflammatory cytokines mediate changes in neurotransmission, in particular on serotonin (5-HT) synthesis and metabolism inducing synaptic plasticity dysfunction [34,35].

Platelets are strictly related to the serotoninergic system and represent specific first-line inflammatory markers. They regulate mechanisms such as endothelial permeability and the recruitment of neutrophils and macrophages. The relationship between platelets and serotonin is bidirectional: on one side, serotonin plays a role in the activation and aggregation of platelets; on the other side, platelets involve a great amount of serotonin in their dense granules and serotonin receptors (5-HT_2A_) and transporters on the cell surface [36]. Interestingly, both platelets and PLR were significantly higher in (hypo)manic than depressed patients, and PLR was regarded as the unique independent predictor of (hypo)mania in our sample. This finding is important, as the activation of platelets, mediated by various inflammatory factors, seems to play an important role in the etiology of psychiatric disorders [37].

Of note, PLR has been recently proposed as a biomarker of high-lethality suicide attempts too [22]. The reduction in brain serotonergic activity accompanied by the up-regulation of some serotonergic post synaptic receptors such as 5-HT_1A_ (expressed mainly in the central nervous system) and 5-HT_2A_ (expressed both centrally and peripherally, such as on platelet surface) are among the most accredited mechanisms associated with the increased risk of attempting suicide [38]. Additionally, high-lethality suicide attempters are characterized by high levels of impulsivity and aggressivity, which represent typical features of mania. In turn, the aggressivity and impulsivity of suicidal attempters seem associated with serotonin content and the total number of 5-HT_2A_ receptors on platelet surface [39]. A recent study showed significantly lower levels of platelet serotonin in suicide attempters compared to non-attempters and high- vs. low-lethality attempters, respectively [40]. Therefore, a strict interplay between platelet levels, serotonin, psychopathological features of (hypo)mania, and suicide in patients with BD could be hypothesized. It would be worth investigating this interaction in BD mixed states.

Even if PLR was regarded as the only independent predictor of (hypo)mania in our sample, it is important to discuss the roles of both NLR and MLR, which appeared increased in (hypo)manic episodes in the univariate analyses. Elevated NLR in BD (hypo)manic episodes may suggest an imbalance in favor of innate immunity, as neutrophils are part of the first line of innate immune defense, while lymphocytes are primarily involved in the adaptive immune response. Of note, NLR was originally developed by intensivists to provide a suitable parameter that could reflect the intensity of stress and/or systemic inflammation in critically ill patients [41], and the cytokine cascade following systemic inflammation appears notably associated with mood disorders [42,43]. It is possible to hypothesize that an elevated neutrophil count and relatively reduced lymphocyte count (2.2 in mania vs. 2.37 × 10^3^/mm^3^ in major depressive episodes) could be due to an inflammatory or chronic stress-induced cellular immunosuppression [44,45,46].

As per the MLR, it has been demonstrated that levels of circulating monocytes are elevated in patients with psychiatric disorders such as BD, major depressive disorder (MDD), and schizophrenia, due to enhanced expression of immune genes and the overproduction of monocytes/macrophage-related cytokines [47]. However, little is known about this index in BD. To our knowledge, only two studies examined MLR in patients with BD [26,27]. Özdin et al. [26] observed higher MLR values in schizophrenic and manic BD patients than healthy controls, while Mazza et al. [27] reported that MLR was significantly higher in the BD manic group than in the BD depressive group. It can be supposed that elevated MLR in BD manic episodes could represent a peripheral marker of brain inflammation, since activation of the microglia may be part of the systemic activation of the mononuclear phagocyte system [47]. However, more studies are needed to better elucidate the role of monocytes in BD mood episodes.

### 4.1. Strengths and Limitations

The present study analyzed the parameters of 294 patients with BD recruited in two Italian university hospitals. To our knowledge, this is the largest cross-sectional study comparing peripheral inflammatory variables in different mood episodes of BD. Nevertheless, some limitations should be discussed. For instance, we did not analyze symptom severity, but only classified patients according to mood episodes, considering only inpatients. Important variables related to the immune system, such as cytokines and CRP, were not included in our analyses. Moreover, concomitant physiological (e.g., menstrual cycle, eating habits) or pathological (e.g., inflammation, cancer, autoimmune disorders, metabolic syndrome) conditions potentially influencing inflammatory status were not considered as covariates in the analysis. Finally, we did not include patients with euthymic BD or healthy controls.

### 4.2. Future Directions

Our findings lay the ground for a novel line of research based on the evaluation of blood cell counts and inflammatory ratios in patients with BD. Inflammatory ratios are in fact low-cost and accessible markers of inflammation, which can be easily derived by a routine blood examination [13]. Moreover, studies have shown significant correlations of inflammatory ratios with other established inflammatory markers like CRP, oxidative stress, and some pro-inflammatory cytokines, both in psychiatric [48,49] and non-psychiatric samples [50,51]. Importantly, these parameters seem less affected by exercise, catecholamine release, and other confounding conditions than single leukocyte parameters or other commonly used markers of inflammation [52].

Several ideas could be implemented by researchers in future studies. First, the systematic administration of validated scales may help to correlate the levels of hematological parameters with the severity of depressive or (hypo)manic symptoms in future studies. Second, it would be worth investigating whether inflammatory ratios follow the same trend of other immunological and inflammatory biomarkers during different BD phases, such as CPR or cytokines (i.e., TNF-α, IL-6, and IL-1). Third, it would be desirable to implement longitudinal studies to examine the intra-subject variability of peripheral blood parameters during different mood episodes. In fact, changes in complete blood count, particularly inflammatory ratios, may anticipate or be concomitant to mood swings in subjects with BD. In this regard, examining the relationship between phenomenology and the biology of BD would be of extreme interest. Fourth, future research should examine the validity of these values in distinguishing manic or depressed states from both euthymic phases and healthy controls, with the proposal of optimal cutoffs. Last, it would be interesting to analyze the potential role of PLR, NLR, and MLR in identifying mixed affective states, which is frequently challenging for clinicians.

## 5. Conclusions

Our findings remark the role of inflammation in the onset and progression of BD, particularly during (hypo)manic episodes. As complete blood count represents a simple and economic examination which is routinely performed in in- and outpatients with BD, investing in this line of research may lead to the discovery of more solid links between the biological parameters and psychopathological variables of BD. The biological phenotyping of such a complex disorder could in turn facilitate the identification of novel therapeutic strategies.

## Figures and Tables

**Figure 1 brainsci-11-00058-f001:**
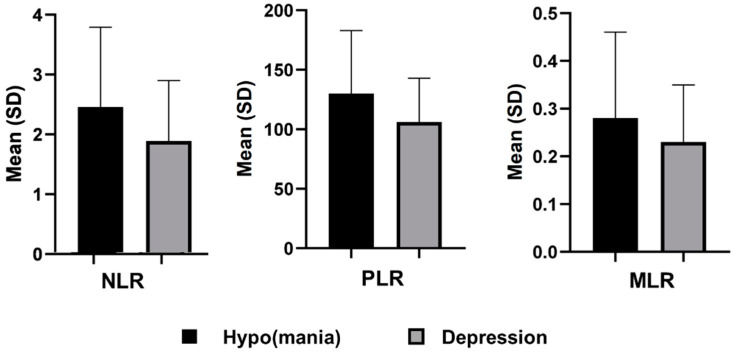
Means and SD of neutrophil-to-lymphocyte ratio (NLR), platelet-to-lymphocyte ratio (PLR), and monocyte-to-lymphocyte ratio (MLR) in bipolar patients experiencing a (hypo)manic and a depressive episode.

**Table 1 brainsci-11-00058-t001:** Characteristics of the sample.

**Socio-Demographic Characteristics (*N* = 294)**	
Gender (female), *N* (%)	157 (53.4)
Age (years), mean ± SD	51.57 ± 13.64
Education level, mean ± SD	11.85 ± 2.05
Marital status, *N* (%)	
Single	145 (49.4)
Married	75 (25.5)
Divorced	51 (17.3)
Widowed	23 (7.8)
Working status, employed *N* (%)	95 (32.3)
Illness phase, *N* (%)	
(Hypo)manic episode	143 (48.6)
Major depressive episode	151 (51.4)
**Pharmacological Treatment (*N* = 236)**	
Antidepressants, *N* (%)	98 (41.5)
Mood stabilizers, *N* (%)	
Valproate	103 (43.6)
Lithium	91 (38.6)
Others	61 (25.8)
Antipsychotics, *N* (%)	202 (85.6)
Typical	45 (19.1)
Atypical	182 (77.1)
Long-acting injection	9 (3.8)
Benzodiazepines, *N* (%)	175 (74.2)
Number of medications, mean ± SD	3.81 ± 1.22

**Table 2 brainsci-11-00058-t002:** Comparison of cell blood count values between (hypo)manic and major depressive episode in bipolar patients.

Mean ± SD	(Hypo)manic Episode (*N* = 143)	Major Depressive Episode (*N* = 151)	*t*	*p*	Cohen’s *d*
Neutrophils	4.87 ± 1.89	4.37 ± 1.56	2.462	0.014 *	0.29
Lymphocytes	2.20 ± 0.78	2.37 ± 0.70	−1.872	0.062	0.23
Monocytes	0.58 ± 0.34	0.53 ± 0.20	1.763	0.079	0.18
Eosinophils	0.21 ± 0.14	0.21 ± 0.13	−0.052	0.959	0
Basophils	0.34 ± 0.02	0.40 ± 0.03	−1.753	0.081	2.35
Platelets	261.25 ± 68.80	234.73 ± 58.97	3.554	<0.001 *	0.41
Neutrophil-to-lymphocyte ratio (NLR)	2.46 ± 1.33	1.89 ± 1.01	3.850	<0.001 *	0.48
Platelet-to-lymphocyte ratio (PLR)	130.13 ± 52.84	106.05 ± 36.79	4.554	<0.001 *	0.53
Monocyte-to-lymphocyte ratio (MLR)	0.28 ± 0.18	0.23 ± 0.12	2.664	0.008 *	0.33
Red blood cell	4.68 ± 0.60	4.58 ± 0.54	1.506	0.133	0.17
Hemoglobin	135.93 ± 17.47	134.60 ± 15.71	0.686	0.493	0.08
Hematocrit	41.10 ± 4.54	40.72 ± 4.34	0.727	0.468	0.08
Mean corpuscular volume	88.34 ± 7.00	89.37 ± 7.62	−1.208	0.228	0.14
Mean corpuscular hemoglobin	29.17 ± 2.72	29.53 ± 2.86	−1.096	0.274	0.13
Mean corpuscular hemoglobin concentration	329.99 ± 12.14	330.17 ± 10.49	−0.136	0.892	0.02
Red blood cell distribution width_coefficient of variation	13.84 ± 1.32	14.02 ± 1.60	−1.052	0.294	0.12

* Statistically significant with *p* < 0.05.

**Table 3 brainsci-11-00058-t003:** Relationship between potential explanatory variables and (hypo)manic episode: results from the logistic regression.

Variables	T	E.S.	Wald	*p*	Exp(B) (95% CI)
Platelet-to-lymphocyte ratio	0.019	0.01	4.292	0.028	1.12 (1.05–1.24)
Neutrophil-to-lymphocyte ratio	−0.52	0.43	1.450	0.229	0.59 (0.26–1.38)
Monocyte-to-lymphocyte ratio	1.63	1.01	2.581	0.108	5.1 (0.699–37.33)
Platelets	−0.001	0.005	0.101	0.750	1.00 (0.99–1.01)
Neutrophils	0.33	0.23	1.938	0.164	1.39 (0.87–2.20)
Constant	−2.65	0.60	19.721	<0.001	0.071

## Data Availability

The data presented in this study are available on request from the corresponding author. The data are not publicly available due to privacy/ethical restrictions.

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
