# Peer review of "Neutrophil-to-Lymphocyte, Platelet-to-Lymphocyte and Monocyte-to-Lymphocyte Ratio in Bipolar Disorder"

_brainsci, 2021, doi:10.3390/brainsci11010058_

Round 1

Reviewer 1 Report

Although this article deserves to be published, important information has been left out by omission, and this should be amended. Since this study has limited scope (as described by the authors in the Strengths and Limitations section), the authors should try make the article as informative as possible.

In order to aid reproducibility for future scientists, the methods section should also describe exactly how these blood cells are obtained, how is the blood drawn, how much blood per patient, the following procedure and how the experimental quantification of all the parameters is performed, i.e. flow cytometry or manual counting in microscope, etc.

Data presentation in table format can be acceptable, but imposes self-inflicted limitations. In table format, the effect size is not immediately visible and instead needs to be calculated by the reader. Also the distribution of data points cannot be displayed and so the signal-to-noise ration is not discernible. The authors should consider improving the reader experience by use of scatter plots laid over bar charts instead, that are presented in a clear and understandable manner to even a non-expert reader. Using charts will make it easier for future scientists to compare their own figures to yours, when they attempt at reproducing the data with their own patient cohort.

Since the articles is effectively based on solely one set of data, this data should be examined further, such as comparing males and females in separate graphs, using raw values that are not corrected. Perhaps it is the case that certain variables better predict the bipolar state in one of the sexes?

Author Response

Q1. Although this article deserves to be published, important information has been left out by omission, and this should be amended. Since this study has limited scope (as described by the authors in the Strengths and Limitations section), the authors should try make the article as informative as possible.

R1. We thank the Reviewer for carefully reading our mansucript. We have tried to reply to all the concerns raised by the Reviewer. We hope that our manuscript has significantly improved after the revision.

Q2. In order to aid reproducibility for future scientists, the methods section should also describe exactly how these blood cells are obtained, how is the blood drawn, how much blood per patient, the following procedure and how the experimental quantification of all the parameters is performed, i.e. flow cytometry or manual counting in microscope, etc.

R2. We thank the Reviewer for the Comment. The methodology has now been clarified in the Methods section (see Paragraph 2.2. “Assessment and Procedures”).

Q3. Data presentation in table format can be acceptable, but imposes self-inflicted limitations. In table format, the effect size is not immediately visible and instead needs to be calculated by the reader. Also the distribution of data points cannot be displayed and so the signal-to-noise ration is not discernible. The authors should consider improving the reader experience by use of scatter plots laid over bar charts instead, that are presented in a clear and understandable manner to even a non-expert reader. Using charts will make it easier for future scientists to compare their own figures to yours, when they attempt at reproducing the data with their own patient cohort.

R3. We thank the Reviewer for the suggestion. We have provided effect sizes (Cohen’s d) in Table 2 with corresponding interpretation in the Results section. Moreover, we added a Figure (Figure 1) including bar charts for the parameters of interest (NLR, PLR, MLR).

Q4. Since the articles is effectively based on solely one set of data, this data should be examined further, such as comparing males and females in separate graphs, using raw values that are not corrected. Perhaps it is the case that certain variables better predict the bipolar state in one of the sexes?

R4. We thank the Reviewer for the comment. Indeed, sex differences in inflammatory ratios represent a novel and interesting research question. In this regard, we have recently submitted a paper to another journal and, therefore, we would prefer not to investigate this aspect in the present article which was designed solely with the purpose of giving a general overview on the topic.

Reviewer 2 Report

The article is interesting and worth publishing. I propose to add a few words in the introduction about neuro-inflammation in mental diseases. It is worth quoting the following works:

10.1016/j.pharep.2019.05.022 

10.1186/1741-7015-10-66 

The article should be reviewed by a native speaker

Author Response

Q1. The article is interesting and worth publishing. I propose to add a few words in the introduction about neuro-inflammation in mental diseases. It is worth quoting the following works:

10.1016/j.pharep.2019.05.022 

10.1186/1741-7015-10-66 

R1. We thank the Reviewer for the comment. We agree with the importance of discussing the role on neuroinflammation in mental disorders and discussed the reference suggested by the Reviewer in the Introduction.

Q2. The article should be reviewed by a native speaker

R2. We thank the Reviewer for the comment. The article has been carefully revised and we are confident that MDPI English editing service will further check for spelling errors and typos before publication.